# Coumarin and Moracin Derivatives from Mulberry Leaves (*Morus alba* L.) with Soluble Epoxide Hydrolase Inhibitory Activity

**DOI:** 10.3390/molecules25173967

**Published:** 2020-08-31

**Authors:** Hong Xu Li, Myungsook Heo, Younghoon Go, Young Soo Kim, Young Ho Kim, Seo Young Yang, Wei Li

**Affiliations:** 1Shenzhen Key Laboratory of Marine Bioresource and Eco-Environmental Science, College of Life Sciences and Oceanography, Shenzhen University, Shenzhen 518060, China; charon0077@gmail.com; 2College of Physics and Optoelectronic Engineering, Shenzhen University, Shenzhen 518060, China; 3College of Pharmacy, Chungnam National University, Daejeon 34134, Korea; inyl1110@naver.com (M.H.); yhk@cnu.ac.kr (Y.H.K.); 4Korean Medicine (KM) Application Center, Korea Institute of Oriental Medicine, Daegu 41062, Korea; gotra827@kiom.re.kr (Y.G.); yskim527@kiom.re.kr (Y.S.K.)

**Keywords:** *Morus alba*, coumarin, moracin, soluble epoxide hydrolase (sEH)

## Abstract

This study identified three coumarins (**1**–**3**), and six moracin derivatives (**4**–**9**). The structures of these natural compounds were determined by the spectroscopic methods, including 1D and 2D NMR methods, and comparison with previous reported data. All of the isolated compounds were assessed for the effects on the soluble epoxide hydrolase (sEH) inhibitory activity. Among them, compounds **1**–**7** exhibited significant inhibitory effect with 100% inhibitory, with IC_50_ values of 6.9, 0.2, 15.9, 1.1, 1.2, 9.9, and 7.7 µM, respectively. A kinetic study revealed that compounds **1**–**4**, and **6** were competitive types of inhibitors, compounds **5** and **7** were mixed types of inhibitors. These results suggest that moracin and coumarin derivatives from mulberry leaves are significant sEH inhibitors.

## 1. Introduction

Inflammation is a normal protective biological response to irritation, injury, or infection. However, appropriate functioning of the immune system is necessary to maintain homeostasis. Prolonged inflammatory response often leads to the onset of chronic diseases, such as cancer, rheumatoid arthritis, and vascular disorders [1]. The inflammatory pathway producing eicosanoids, eicosanoids are a group of lipid mediators generated from arachidonic acid (ARA) by activity of cyclooxygenases (COX), lipoxygenases (LOX), and cytochrome P450 (CYP450) enzymes [2], i.e., signaling molecules derived from arachidonic acid, has been implicated in a variety of disorders, including stroke, hypertension, and renal diseases [3]. The pathway is mediated by cytochrome P450 enzymes and results in the production of two types of compounds, namely hydroxyeicosatetraenoic acids (HETEs) and epoxyeicosatrienoic acid isomers (EETs) [3]. Notably, soluble epoxide hydrolase (sEH) converts EETs to their corresponding diols (i.e., dihydroxyeicosatrienoic acids, DHETs), lead to reduce effects of EETs on the cardiovascular system through vasodilation, antimigration of vascular smooth muscle cells, and anti-inflammatory responses. Thus, sEH is considered a potential therapeutic target for the treatment of vascular diseases [4]. 

Morus, a genus of flowering plants in the family Moraceae, comprises 10–16 species of deciduous trees commonly known as mulberries. In this study, we mainly studied two types of compounds—coumarin and moracin. Coumarin belong to the benzopyrone type of compounds. The analogues of coumarin consist of various substances of phenolic class types. Biosynthesis of coumarins in plants follow the phenylpropanoid pathway [5]. Moreover, the distinctive and adaptable oxygen containing heterocyclic structure declared such an importance scaffold in coumarin compounds upon medicinal chemistry [6]. In the past decades, numerous derivatives of coumarins have been used as anticoagulant agents due to their resemblance to Vitamin K. In addition, coumarin analogues have been reported as inhibitors of sEH in previously reported literature [7], as well as many other inhibitor agents. The root bark, stem bark, and leaves of *Morus alba*, *M. lhou*, *Morus macroura* are the main sources for aryl-benzofuran derivatives, including the moracins. A large volume of research has been carried out on moracins and their derivatives, which has shown the pharmacological importance of this benzofuran heterocyclic nucleus. *Morus alba* L. (Moraceae) is widely cultivated in Asia and has been utilized in traditional medicine for decades. The *M. alba* herb is used to treat diabetes, inflammation, and obesity [8]. The benzofuran heterocycles are fundamental structural units in a wide range of biologically active natural products as well as synthetic materials. Moracin family is biologically active natural products containing benzofuran heterocycle as basic structural units. It has been shown that aryl-benzofurans isolated from this plant exhibit significant inhibitory activity against nitric oxide production [9]. Moreover, our previous studies indicated that several aryl benzofuran and flavonol derivatives displayed strong activity in the treatment of obesity and melanogenesis [10,11]. Hence, *M. alba* is a potential source of numerous natural products with important biological activities.

## 2. Results and Discussion

### 2.1. Isolation and Structural Elucidation

In the present study, nine compounds were isolated from the MeOH extract of *M. alba* (Figure 1). The structures of the compounds were determined by various spectroscopic methods, including 1D and 2D nuclear magnetic resonance to give aesculetin (**1**) [12], scopoletin (**2**) [13], scopoline (**3**) [14], moracin B (**4**) [15], moracin J (**5**) [16], moracin M (**6**) [17], moracin M 3’-*O*-*β*-glucopyranoside (**7**) [18], moracin M 6-*β*-D-glucopyranoside (**8**) [19], and mulberroside F (**9**) [20] (See Appendix A).

### 2.2. Bioassays

Over the last 20 years, sEH has been linked to numerous pathological conditions, including cardiovascular and neurological diseases [3]. Additionally, its role in the central nervous system disorders has also been established. Thus, inhibition of this enzyme shows robust therapeutic potential. In the present study, the candidate inhibitory compounds **1**–**7** were subjected to an enzyme kinetics evaluation to access the binding mode between the receptor and ligands. The enzyme inhibition properties of the derivatives were modeled using double-reciprocal plots (Lineweaver‒Burk and Dixon analyses). It was determined that compounds **1**–**4** and **6** were competitive inhibitors. The analysis of the Lineweaver–Burk plot suggested that increasing the inhibitor concentration increased the *K*_m_ values without affecting *V_m_* [21]. Moreover, the *K*_i_ values for compounds **1**–**4** and **6** were calculated from the Dixon plots and were equal to 1.2, 0.3, 5.4, 1.0, and 1.5 µM, respectively. Compounds **5** and **7** were found to be mixed inhibitors. Analysis of the Lineweaver–Burk plot indicated that increasing the inhibitor concentration increased *K*_m_, but decreased the *V*_m_ values [21]. The *K_i_* values for compounds **5** and **7** were also calculated from the Dixon at 2.1 and 5.8 µM, respectively (Figure 2).

The sEH inhibitory effects of compounds **1**‒**9** isolated from *M. alba* were subsequently investigated using recombinant human sEH incubated in the presence of PHOME, which is an artificial substrate for fluorescence detection (Table 1). All of the isolated derivatives were tested in 100 µM solutions against the enzyme. Notably, compounds **1**‒**7** exhibited as 100% inhibitory activity against sEH, while analogs **8** and **9** displayed insignificant effects (<50%). In the past, phytochemistry and bioactivity studies primarily focused on aryl benzofuran derivatives [22]. The sEH inhibitory activity of coumarins established in the present work provides a valuable platform for further bioactivity evaluation. It is noteworthy that coumarin analogs have low molecular weights and show high degree of lipid solubility, facilitating transmembrane diffusion [23]. We determined that compound **1** had a lower IC_50_ value (6.9 µM) than derivative **3** (15.9 µM). Moreover, derivative **2** not only displayed robust sEH inhibitory effects, but also exhibited the lowest IC_50_ value (0.2 µM) out of all nine isolated compounds. The presence of three types of functional groups in the molecules, specifically –OH, –OCH_3_, and –OGlc, particularly drew our attention. Both the inhibitory effects and the IC_50_ values were considerably affected by different functional groups. Replacing the –OH moiety at the C-6 position in compound **1** with a –OCH_3_ group led to a 34-fold decrease in the IC_50_ value than before. On the other hand, the presence of some functional groups, e.g., –OGlc, resulted in an increase in the IC_50_ value. Similarly, to the coumarin derivatives, the moracin analogs contain the same three types of functional groups (i.e., –OH, –OCH_3_, and –OGlc). Hence, the structural properties and the determined sEH inhibitory effects of compounds **1**‒**9** allowed us to investigate the structure-activity relationship (Figure 3).

Based on the exhibited inhibitory effects, the aryl benzofurans could be divided into three categories. The first category included compounds **4** and **5**, while the second, derivatives **6** and **7**. All compounds in this group displayed inhibitory activity of = 100% with IC_50_ values of 1.1, 1.2, 9.9, and 7.7 µM, respectively. The last category included derivatives **8** and **9** with low inhibitory activities of 18.3% and 17.1%, respectively. The classification was not only based on the IC_50_ values, but also on the presence of specific functional groups. Compounds **7**, **8** and **9** all contain a –OGlc functional group; however, they display various inhibitory effects and IC_50_ values. It was speculated that the dissimilarities were a consequence of different functionalities on the A or B ring in the structures. The obtained results suggested that in the case of moracin compounds, a –OGlc moiety on the A ring decreased the sEH inhibitory effect. However, the presence of this functionality on the B ring had a minor effect on the inhibitory activity. The difference between the structures of isomers 4 and 5, which possess identical B rings, is the position of the substituents at the C-5 and C-6 positions. The comparison of the data for these compounds revealed that the –OCH_3_ group plays a significant role in sEH inhibition, resulting in both an increase in the inhibitory effects and a decrease in the IC_50_ values. Furthermore, for compounds **6** and **7**, the replacement of the –OH moiety with the –OCH_3_ group led to an increase in the IC_50_ values. Overall, it was established that the –OCH_3_ functionality increased the sEH inhibitory effects most significantly, followed by the –OH group. Moreover, the presence of the –OGlc moiety resulted in an increase in the IC_50_ values for some compounds or a decrease in the inhibitory effects for other derivatives.

Previous studies have reported a crystal structure showing the interaction between sEH and its potent inhibitor 3-phenylglutaric acid (Protein Data Bank (PDB) code: 3ANS) [24,25]. Based on this information, we investigated, the binding interactions of compound 2, 4, and 5, which were effective on the sEH inhibition, with amino acid residues in sEH by protein–ligand docking simulation using AutoDock Vina and LigPlot+ software (Figure 4). Molecular docking simulation indicated that sEH may interact with compound 2, 4, and 5 by forming several hydrogen bonds and hydrophobic interactions (Figure 4 and Table 2). The pharmacophore analysis suggested that compound 2 created a hydrophobic interaction and a hydrogen bond with the two amino acids Asp335 and Trp466 [7], respectively, among the sEH catalytic triads (Asp335, Tyr383, and Trp466), and a strong π-π interaction with Trp336, such like 3-phenylglutaric acid [24,25]. While compound 4 and 5 seem to block the catalytic pocket of sEH by the interaction of A ring with all catalytic triads and C ring with Ser407, Leu408, Ser415, Leu417, and Met419, locating on the opposite side of Trp336. These binding sites suggest that these amino acid sequences are crucial receptors in the inhibition of sEH enzyme activity. The interaction with catalytic triads were known to conservatively contribute to stabilizing the binding between sEH and its various inhibitors [24].

## 3. Conclusions

In the present study, nine compounds (**1**–**9**) were isolated from the MeOH extract of *M. alba*. The analysis of the sEH inhibitory effects indicated that coumarin and aryl benzofuran derivatives show potential biological activities. Inhibitory activity of =100% was noted for some compounds, showing the potential of coumarins and aryl benzofurans for the treatment of inflammatory disorders. Notably, a remarkably low IC_50_ value was determined in the case of compound **2** (0.2 µM). Nonetheless, further research is necessary to confirm compounds **1**–**7** as potential drug candidates for the treatment of inflammatory diseases. Therefore, we identified some bioactive compounds corresponding to the traditional treatment usage, which might prove by in vitro methods, molecular docking simulation, and pharmacophore analysis. As a rich natural product resource of Moraceae family, it is important to study some compounds which might collaborate working with each other and apply a better choice for the patients, especially with some chronical physical disorders. Thus, a low cost and toxicity treatment strategy could provide for more needed people.

## 4. Materials and Methods

### 4.1. General Information

Optical rotations were determined using a Jasco DIP-370 automatic polarimeter. The NMR spectra were recorded using a JEOL ECA 600 spectrometer (^1^H, 600 MHz; ^13^C, 150 MHz), The LCQ advantage trap mass spectrometer (Thermo Finnigan, San Jose, CA, U.S.A.) was equipped with an electrospray ionization (ESI) source, and High-resolution electrospray ionization mass spectra (HR-ESI-MS) were obtained using an Agilent 6530 Accurate-Mass Q-TOF LC/MS system. Preparative HPLC was performed using a GILSON 321 pump, 151 UV/VIS detector (Gilson, VILLIERS-LE-BEL, France), and RStech HECTOR-M C_18_ column (5-micron, 250 × 21.2 mm) (RS Tech Crop, Chungju, South Korea). Column chromatography was performed using a silica gel (Kieselgel 60, 70-230, and 230-400 mesh, Merck, Darmstadt, Germany), YMC RP-18 resins, and thin layer chromatography (TLC) was performed using pre-coated silica-gel 60 F_254_ and RP-18 F_254_S plates (both 0.25 mm, Merck, Darmstadt, Germany).

### 4.2. Plant Material

Dried leaves of *Morus alba* L. were purchased from herbal company, Naemome Dah, Ulsan, Korea, in September 2015. Its scientific name was identified by one of author (Prof. Young Ho Kim). A voucher specimen (CNU 16004-1) was deposited at the Herbarium of College of Pharmacy, Chungnam National University, Republic of Korea.

### 4.3. Extraction and Isolation

The dried leaves of *M. alba* (2.9 kg) was refluxing extraction with MeOH (10 L × 3) times. The total extraction (384.0 g) of MeOH was suspended in deionized water and partitioned with n-hexane, yielding n-hexane fraction (1A, 166.0 g) and water fraction. Then the water fraction was partitioned sequential with EtOAc and n-BuOH, yielding EtOAc fraction (1B, 16.1 g), *n*-BuOH fraction (1C, 65.0 g) and water fraction (1D, 94.0 g). The EtOAc fraction was subjected to a silica gel column chromatography with a gradient of CHCl_3_: MeOH: water (20:1:0, 15:1:0, 10:1:0, 8:1:0, 6:1:0.1, 4:1:0.1, 2:1:0.1, and 100% MeOH) to give 8 fractions (1B-1–1B-8). The fraction 1B-1 was performed separation with a gradient of MeOH: water (1:4, 1:3, 1:2, 1:1, and MeOH) by middle pressure liquid chromatography (MPLC) using C_18_ column to give 5 fractions (1B-1-1–1B-1-4). Subfraction 1B-1-2 was separated by a Sephadex LH-20 column and eluted by MeOH and its subfractions were isolated by prep-HPLC to give compounds **1** (3.2 mg) and 2 (10.1 mg). Subfraction 1B-1-4 was separated by a Sephadex LH-20 column and eluted by MeOH and its subfraction was isolated by prep-HPLC to give compound **4** (6.1 mg). The fraction 1B-4 was isolated with a gradient of MeOH: water (1:4, 1:3, 1:2, 1:1, and MeOH) by MPLC using C_18_ column to give 3 fractions (1B-4-1–1B-4-3). The fraction 1B-5 was isolated with a gradient of MeOH: water (1:3, 1:2, 1:1, and MeOH) by MPLC using C_18_ column to give 4 fractions (1B-5-1–1B-5-4). Subfraction 1B-5-2 was separated by a Sephadex LH-20 column and eluted by MeOH and its subfraction was isolated by prep-HPLC to give compound **5** (6.5 mg). The fraction 1B-8 was isolated with a gradient of MeOH: water (1:2, 1:1, and MeOH) by MPLC using C_18_ column to give 9 fractions (1B-8-1–1B-8-9). Subfraction 1B-8-5 was separated by a Sephadex LH-20 column and eluted by MeOH and its subfraction was isolated by prep-HPLC to give compound **3** (1.1 mg). Subfraction 1B-8-7 was separated by a Sephadex LH-20 column and eluted by MeOH and its subfraction was isolated by prep-HPLC to give compound **6** (20.1 mg). Subfraction 1B-8-8 was separated by a Sephadex LH-20 column and eluted by MeOH and its subfraction was isolated by prep-HPLC to give compound **8** (9.7 mg). Subfraction 1B-8-9 was separated by a Sephadex LH-20 column and eluted by MeOH and its subfraction was isolated by prep-HPLC to give compound **7** (6.1 mg). The water fraction was subjected on a HP-20 column, and eluted with water, 25% MeOH, 50% MeOH, 75% MeOH, and 100% MeOH, yield 5 fractions (1D-1–1D-5). Fraction 1D-2 and 1D-3 were combined (1D-2-1), and isolated with a gradient of MeOH: water (1:4, 1:3, 1:2, 1:1, and MeOH) by MPLC using C_18_ column to give 5 fractions (1D-2-1-1–1D-2-1-5). Subfraction 1D-2-1-2 was separated by a Sephadex LH-20 column and eluted by MeOH and its subfraction was isolated by prep-HPLC to give compound **9** (17.8 mg).

### 4.4. sEH Assay

The soluble epoxide hydrolase (sEH) assay, bis-Tris methane (B9754), and albumin (A8806) were purchased from Sigma Aldrich (St. Louis, MO, USA). Human recombinant soluble epoxide hydrolases (sEH, 10011669), and 3-phenyl-cyano(6-methoxy-2-naphthalenyl)methyl ester-2-oxiraneacetic acid (PHOME) (10009134) were purchased from the Cayman Chemical Company (Cayman, MI, USA). The 96-well white plate was purchased from Costar (Corning, NY, USA). The fluorescence intensity measurements were conducted utilizing the Tecan infinite F200 microplate reader (Tecan, Mannedorf, Switzerland).

The enzymatic assays were carried out according to previously reported methods, with some modifications [3]. A 130 µL aliquot of recombinant human sEH (12.15 ng/mL) was diluted with the buffer (25 mM bis-Tris-HCl containing 0.1 mg/mL BSA, pH 7.0). Subsequently, 20 µL of MeOH and 50 µL of PHOME (10 µM) were added. The amount of the substrate converted to the product by the enzyme was measured by fluorescence photometry (330 nm excitation filter and 465 nm emission filter), according to the following equation:Enzyme activity (%) = [S_40_ − S_0_/C_40_ − C_0_] × 100
where C_40_ and S_40_ are the fluorescence of the control and inhibitor after 40 min, while S_0_ and C_0_ indicate the fluorescence of the inhibitor and control at 0 min, respectively. In the study, 12-(3-adamantan-1-yl-ureido)-dodecanoic acid (AUDA) was employed as a positive control, and 10% of MeOH was used as blank control. The IC_50_ values were measured according to the concentration over 50% of inhibition ratio. Then various concentrations of substrate were diluted in orders to calculate IC_50_ values using Hyperbola, single rectangular formula y = ax/(b + x) to yield coefficient standard error, a and b, IC_50_ = 50 × b/a − 50.

### 4.5. sEH Kinetic Assay

Kinetic assays were carried out under steady-state conditions. The enzyme inhibition properties of the components were modeled using double-reciprocal plots (Lineweaver–Burk and Dixon analyses). Briefly, 50 µL of sEH and 20 µL of various concentrations of the analyzed compounds in MeOH were added into each well of a 96-well plate. 80 µL of a 25 mM bis-Tris-HCl buffer (pH 7.0) containing 0.1% BSA and 50 µL of the PHOME substrate (5–80 µM) were then added into each well. The enzymatic reaction was initiated at 37 °C and the formation of the products resulting from the hydrolysis of the substrates was monitored over 30 min at excitation and emission of 330 and 465 nm, respectively [26].

### 4.6. Molecular Docking Simulation and Pharmacophore Analysis

The compounds **2**, **4**, and **5** were docked onto the catalytic pocket of sEH retrieved from the Protein Data Bank (www.rcsb.org, PDB code: 3ANS) [24], using AutoDock Vina integrated with UCSF Chimera v1.14 [27]. Subsequently, the interaction between sEH and each compound was analyzed based on the docking simulation result using LigPlot+ v1.4.5 [28]. Amino acid residues involved in the interactions were indicated with red (hydrophobic interactions) and green (H-bonds).

## Figures and Tables

**Figure 1 molecules-25-03967-f001:**
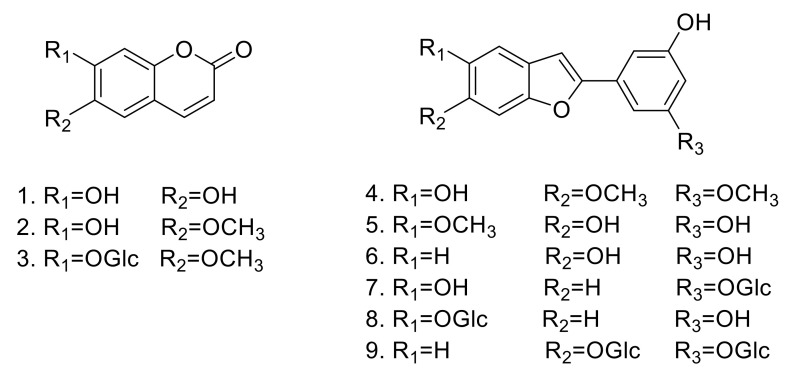
Structures of compounds **1**–**9** isolated from *M. alba*.

**Figure 2 molecules-25-03967-f002:**
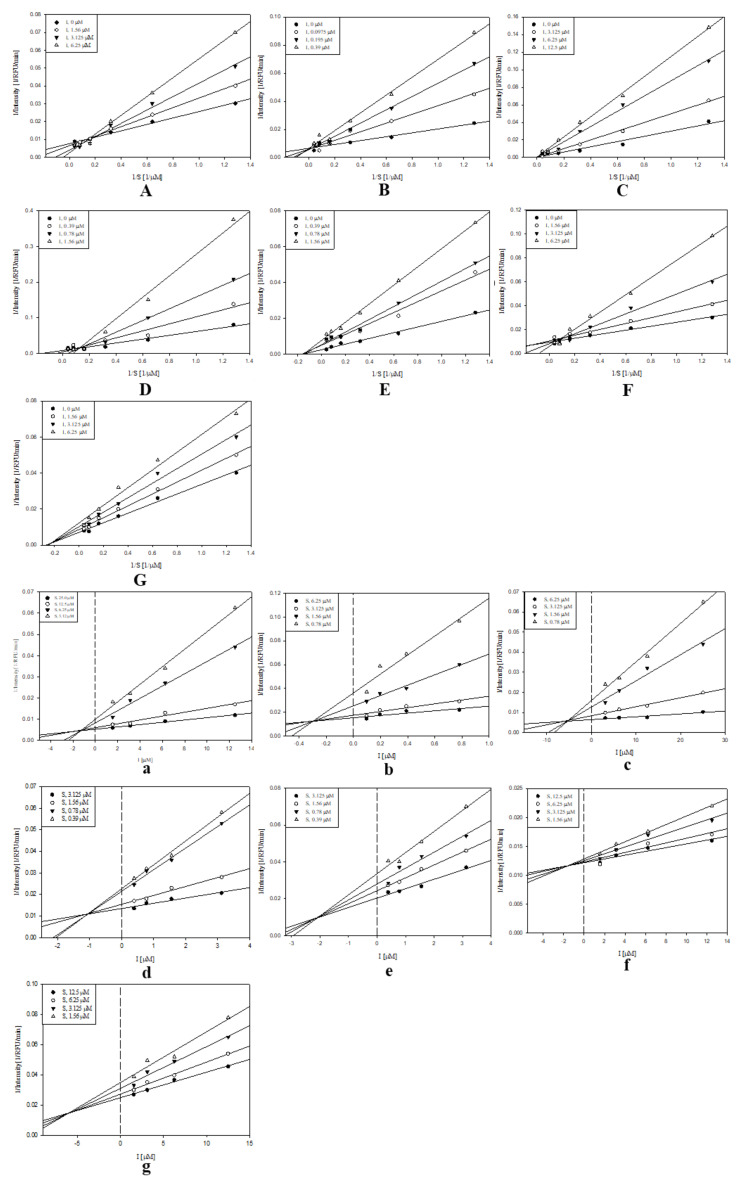
Study of the binding mechanisms between compounds **1**–**7** and sEH: (**A**–**G**) Lineweaver−Burk plots for compounds **1**–**7**, respectively; (**a**–**g**) Dixon plots for compounds **1**–**7**, respectively. Data are the mean of three experiments carried out in triplicate and were determined by one-way analysis of variance, followed by Dunnett’s multiple comparison test, *p* < 0.05 versus control.

**Figure 3 molecules-25-03967-f003:**
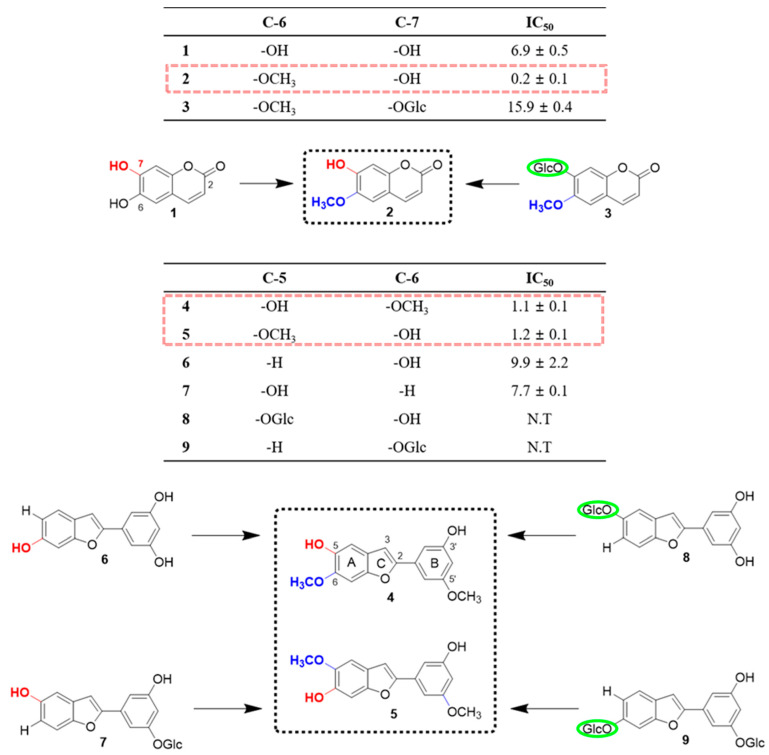
Identification of the structure-activity relationship based on the soluble epoxide hydrolase (she) inhibitory effects of compounds isolated from the leaves of *M. alba*.

**Figure 4 molecules-25-03967-f004:**
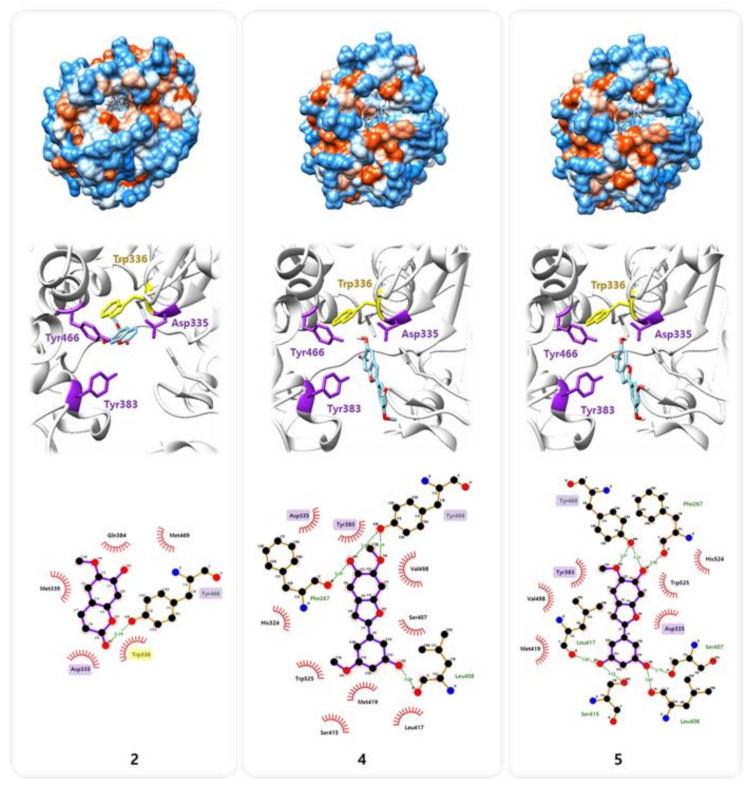
Molecular docking simulation of compounds **2**, **4**, and **5** into the predicted binding site of sEH.

**Table 1 molecules-25-03967-t001:** Inhibitory effects of isolated compounds **1**–**9**.

Inhibition of Compounds on sEH			
Compounds	100 µM(%)	IC_50_ (µM)	Type (*Ki*, µM)
**1**	=100	6.9 ± 0.5	competitive (1.2 ± 0.4)
**2**	=100	0.2 ± 0.1	competitive (0.3 ± 0.1)
**3**	=100	15.9 ± 0.4	competitive (5.4 ± 0.7)
**4**	=100	1.1 ± 0.1	competitive (1.0 ± 0.3)
**5**	=100	1.2 ± 0.1	mixed (2.1 ± 0.6)
**6**	=100	9.9 ± 2.2	competitive (1.5 ± 0.2)
**7**	=100	7.7 ± 0.1	mixed (5.8 ± 0.1)
**8**	18.3 ± 4.2	N.T ^b^	N.T
**9**	17.1 ± 3.3	N.T	N.T
AUDA ^a^		11.6 ± 0.3 (nM)	

sEH activity was expressed as the percentage of control activity. Values represent means ± SD (*n* = 3). ^a^ Positive control. ^b^ N.T: Not Tested.

**Table 2 molecules-25-03967-t002:** Pharmacophore analysis between sEH and compounds **2**, **4**, and **5.**

Compounds	Receptor ^a^	
	Hydrogen Bonds (Å)	Hydrophobic Interactions
**2**	Y466 (3.18)	D335, W336, M339, Q384, M469
**4**	F267 (3.05), L408 (3.04), Y466 (2.88, 3.10)	D335, Y383, S407, S415, L417, M419, V498, H524, W525
**5**	F267 (2.90), L408 (2.83), S407 (2.75), S415 (3.13), L417 (3.03), Y466 (2.97, 3.13)	D335, Y383, M419, V498, H524, W525

^a^ Amino acid sequence number of receptors.

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
