# Peer review of "Coumarin and Moracin Derivatives from Mulberry Leaves (*Morus alba* L.) with Soluble Epoxide Hydrolase Inhibitory Activity"

_molecules, 2020, doi:10.3390/molecules25173967_

Round 1

Reviewer 1 Report

Response 1: The supplementary data was added to prove the purity of these compounds.

->This reviewer cannot find the supplementary data.

Response 2: In this study, for the statistical analysis we used the average values of all the experiments to minimal the disturbance and derivation.

->This reviewer cannot understand this response. Although the authors mentioned one-way ANOVA and Dunnett's test were performed in the legend of Figure 2, this reviewer cannot tell the difference between significant and insignificant data. In addition, there is no description about the statistical analysis in Results and Discussion section.

After revision of the manuscript, the sEH inhibition at 100 µM has been changed to 100% at the highest in Table 1. So, the description "… over 100% inhibitory, …" in the abstract should be revised.

Author Response

1: The supplementary data was added to prove the purity of these compounds.
This reviewer cannot find the supplementary data.

Response 1: The supplementary data was added and enclosed in the manuscript.

2: In this study, for the statistical analysis we used the average values of all the experiments to minimal the disturbance and derivation.
This reviewer cannot understand this response. Although the authors mentioned one-way ANOVA and Dunnett's test were performed in the legend of Figure 2, this reviewer cannot tell the difference between significant and insignificant data. In addition, there is no description about the statistical analysis in Results and Discussion section.

Response 2: The dates showed in average values and standard deviations were added, in order to measure the amount of variation or dispersion of all the parallel experiments. For figure 2, A – G were used to determine the types of inhibition, a – g were used to calculate the Ki values. During the experiments, data are the mean of three experiments carried out in triplicate and were determined by one-way analysis of variance, followed by Dunnett’s multiple comparison test was calculated along with standard deviation and confidence interval. In results and discussion section, Lineweaver‒Burk and Dixon analyses were mentioned in order to give the statistical results.

3. After revision of the manuscript, the sEH inhibition at 100 µM has been changed to 100% at the highest in Table 1. So, the description "… over 100% inhibitory, …" in the abstract should be revised.

Response 3: This comment was corrected in the manuscript.

Reviewer 2 Report

The authors reported the identification of three coumarins (1–3), and six moracin derivatives (4–9). The structures of these natural compounds were determined by the spectroscopic methods including 1D and 2D NMR methods and comparison with previous reported data. The manuscript is very interesting and is well written. It is a very good presented and the conclusion supported by results. The manuscript can be accept in present form. 

Author Response

Response : Thank you for your consideration we were much appreciated for your advice.

Reviewer 3 Report

The paper by Li et al. is an interesting study about the inhibitory effect of coumarins and muracin derivatives isolated from mullberry leaves on sEH activity.

Coumarin derivatives have been previously reported as inhibitors of sEH and this study is citied by authors. However the effect of muracin derivatives on sEH is the novelty in present paper.

Authors showed an extensive data about sEH inhibition effects including structure-activity relationships of isolated compounds. While the results are well presented, there are a few points of revision:

  1. I am confused about the methodology used for identification of isolated compounds. Authors stated in the abstract that the structures of isolated compounds were determined by the spectroscopic methods including 1D and 2D NMR methods. However in methods section they described in details LC-MS systems with Thermo ion trap and Agilent Q-TOF high resolution mass specs equipped with ESI ionisation. It should be clarified which analytical technique authors have used for identification? MS or NMR or both? It would be nice to spend more words on the identification and provide more details about the analysis, especially since authors investigated structure-activity relationships of isolated compounds.
  2. All of the isolated derivatives were tested in 100 μM solutions against the enzyme. Did authors performed any preliminary studies about the concentration of these derivatives?
  3. The protocol how the IC50 was calculated should be described in details in methodology section
  4. sEH is considered as a potential therapeutic target for the treatment of vascular diseases. From the point of view of treatment, would the isolated derivatives be given to the patients by oral administration for instance? How the authors envisage a treatment? Can they comment on that?

Author Response

The paper by Li et al. is an interesting study about the inhibitory effect of coumarins and moracin derivatives isolated from mulberry leaves on sEH activity.
Coumarin derivatives have been previously reported as inhibitors of sEH and this study is citied by authors. However, the effect of moracin derivatives on sEH is the novelty in present paper.
Authors showed an extensive data about sEH inhibition effects including structure-activity relationships of isolated compounds. While the results are well presented, there are a few points of revision:

1. I am confused about the methodology used for identification of isolated compounds. Authors stated in the abstract that the structures of isolated compounds were determined by the spectroscopic methods including 1D and 2D NMR methods. However, in methods section they described in details LC-MS systems with Thermo ion trap and Agilent Q-TOF high resolution mass specs equipped with ESI ionisation. It should be clarified which analytical technique authors have used for identification? MS or NMR or both? It would be nice to spend more words on the identification and provide more details about the analysis, especially since authors investigated structure-activity relationships of isolated compounds.

Response 1: For this study, some compounds were not showed in this manuscript. From our early published paper, reference 10, aminicotin (compound 16) was previously isolated and identified by both NMR and LC-MS methods. Therefore, we mentioned these methods in this part.

2. All of the isolated derivatives were tested in 100 μM solutions against the enzyme. Did authors perform any preliminary studies about the concentration of these derivatives?

Response 2: All the isolated compounds were tested under 100 µM/ml in order to screen for the bioactive inhibitors initially. Then, these inhibition rate over 50% compounds were conducted for the further study. Our lab set up this method and considered enzyme dynamic velocity that 100 µM/ml might be a suitable concentration for most compounds to screen for bioactive inhibitors. 

3. The protocol how the IC50 was calculated should be described in detail in methodology section

Response 3: The IC50 values were measured according to the concentration over 50% of inhibition ratio. Then various concentrations of substrate were diluted in orders to calculate IC50 values using Hyperbola, single rectangular formula y=ax/(b+x)  to yield coefficient standard error, a and b, IC50 = 50×b∕a-50 .

4. sEH is considered as a potential therapeutic target for the treatment of vascular diseases. From the point of view of treatment, would the isolated derivatives be given to the patients by oral administration for instance? How the authors envisage a treatment? Can they comment on that?

Response 4: We hope we could conduct these experiments in the future. However, there are many works to be done and many aspects to be considered. For instance, what is the mechanism for some of these compounds working in one certain pathway or several pathways.